# Aqueous extracts of *Urtica dioica* (stinging nettle) leaf contain a P2-purinoceptor antagonist—Implications for male fertility

**Nicole T. Eise[1,2], Jamie S. Simpson[2], Philip E. Thompson[2], Sabatino Ventura[1]***

**1** Drug Discovery Biology, Monash Institute of Pharmaceutical Sciences, Monash University, Parkville, Australia, **2** Medicinal Chemistry, Monash Institute of Pharmaceutical Sciences, Monash University, Parkville, Australia

\* sab.ventura@monash.edu

**Data Availability Statement:** All relevant data are within the paper.

**Funding:** The authors received no specific funding for this work.

## Abstract

Stinging nettle root and leaf extracts were tested for their effect on prostatic smooth muscle contractility. Root extract did not affect electrical field stimulation induced-nerve mediated contractions of isolated rat prostates. On the other hand, leaf extract attenuated electrical field stimulation-induced contractions at all frequencies. Similarly, contractions elicited by exogenous administration of ATP and αβ-methylene ATP were inhibited by leaf extract, whereas contractions elicited by exogenous administration of noradrenaline or acetylcholine were unaffected. The active component was present within the aqueous phase of the leaf extract. In mouse mating studies, stinging nettle leaf extract (50 mg p.o. daily) reduced male fertility by 53% compared to vehicle-treated male mice. Cardiovascular parameters were unaffected by administration of stinging nettle leaf extract ($p \geq 0.057$). Treated mice exhibited normal mating behaviour. Bladder and testes weighed less in stinging nettle leaf extract treated mice. All other organs and total body weight were unaffected. It is concluded that stinging nettle leaf extract reduces contractility of genitourinary smooth muscle by acting as an antagonist at postjunctional P2X1-purinoceptors. These data indicates that blocking sperm transport through pharmacological blockade of P2X1-purinoceptors via oral administration is consistent with an effective and convenient biological strategy male contraception.

## Introduction

Historically, phytotherapeutics have been used widely by many cultures for various ailments including lower urinary tract symptoms (LUTS) [1]. Traditional medicine uses the roots of the stinging nettle plant for the treatment of urinary problems including prostatitis, nocturia, frequency, and dysuria [2, 3]. Consequently, the root extract is the part of the plant found in commercially available preparations containing stinging nettle for the symptomatic treatment of lower urinary tract symptom associated with benign prostatic hyperplasia (BPH). In these commercially available supplements, stinging nettle root extract is used either as a single agent [4] or in combination with other plant extracts such as saw palmetto [5].

**Competing interests:** The authors have declared that no competing interests exist.

Although not commercially used to treat LUTS, stinging nettle leaf is used in traditional medicine to brew teas that are used to increase diuresis and treat LUTS [1, 6, 7]. Specifically, the aerial parts of the plant are traditionally decocted as a tea in Turkey for the treatment of nocturia [1], as well as in the alpine regions of Italy, to increase urinary output [6]. In addition, the leaves have also been used in traditional and folk medicines to lower blood pressure for the treatment of hypertension [2, 8–12]. In agreement, aqueous extracts of the aerial parts of stinging nettle have been demonstrated to produce an acute dose-dependent reduction in arterial blood pressure in rats [13, 14]. Similarly, aqueous extracts of the aerial parts of stinging nettle have been reported to stimulate non-cholinergic and non-adrenergic pathways to mediate bradycardia in isolated rat hearts [15].

Increased urinary flow, particularly in elderly men, can be facilitated by relaxation of prostatic tissue surrounding the urethra thereby removing obstruction produced by the enlarged prostate. This is the quickest and most effective therapeutic mechanism for relieving LUTS in men [16]. Similarly, a reduction in blood pressure suggests a relaxant effect on vascular smooth muscle. Both prostatic and vascular smooth muscle is similarly innervated by sympathetic nerves that release noradrenaline and ATP.

The traditional use of stinging nettle leaf extracts implies that they have a smooth muscle relaxant effect. Accordingly, stinging nettle root extract has been reported to possess vasorelaxant activity mediated by both endothelial nitric oxide release and a direct negative inotropic action [17]. In addition, a reduction in blood pressure in spontaneously hypertensive Wistar rats was observed after the administration of either the flavonoid quercetin [18, 19] or chlorogenic acid [20], both of which have been identified in stinging nettle plants [20–24] by phytochemical analysis. Quercetin has also demonstrated vasorelaxant activity on aortic preparations pre-contracted with KCl or noradrenaline possibly by the inhibition of protein kinase C and/or a decrease in $Ca^{2+}$ uptake [25].

Despite these pharmacological investigations into the mechanisms of action of stinging nettle, a knowledge gap remains in the definitive understanding of its pharmacological mechanism of action, as the majority of evidence is anecdotal with minimal robust studies and clinical trials having been conducted. Therefore, one of the aims of this research was to investigate the activity of a commercially available stinging nettle extract on the contractility of the genitourinary smooth muscle of the rat prostate gland and vas deferens, and to determine the mechanism of action.

Although a specific pharmacological mechanism for either stinging nettle root or leaf extract on smooth muscle relaxation has not yet been identified, the potential smooth muscle relaxant effect of these extracts may translate to therapeutic benefits in either genitourinary or vascular disorders. For instance, prostatic smooth muscle relaxation is an effective means of alleviating male LUTS associated with BPH [26], while pharmacological inhibition of vas deferens contractility has been proposed as a biological strategy for male contraception [27]. This study investigated the effects of both stinging nettle root and leaf extracts on rat prostatic smooth muscle contractility *in vitro* using isolated tissue experiments to identify a mechanism of action. A follow up *in vivo* study was also conducted in mice to assess effects on male fertility. Liquid-liquid partitioning was employed as an initial step toward isolating the active component.

## Materials and methods

### Animals

Male Sprague-Dawley rats were obtained at 7–9 weeks of age and male wild-type C57Bl/6 mice were obtained at 7–8 weeks of age from the Monash Animal Research Platform (MARP)

(Monash University, Clayton). P2X1-purinoceptor knockout C57Bl/6 mice were bred internally from heterozygous breeding pairs of P2X1-purinoceptor knockout mice originally obtained from Prof R.J. Evans (Department of Cell Physiology & Pharmacology, University of Leicester, UK). All animals were exposed to a photoperiod of 12 hours light and 12 hours dark and housed under standard conditions at 22˚C; food and water were accessed *ad libitum*. To obtain tissues, animals were euthanised by asphyxiation through exposure to $CO_2$ gas. Prior approval for animal experimentation was obtained from the Monash University Standing Committee of Animal Ethics in Animal Experimentation; ethics numbers VCPA 2009/15 and MIPS 2013/15 for the use of Sprague-Dawley rats, MIPS 2014–04 and MIPS 2015–06 for the use of genetically modified and wild-type C57Bl/6 mice, respectively. All experiments were performed in accordance with relevant guidelines and regulations and all methods are reported in accordance with ARRIVE guidelines [28]. Polymerase chain reaction (PCR) was used to routinely determine the genotype of individual mice in the P2X1-purinoceptor knockout mouse breeding colony, as previously described [27].

## Rat dissection

Rats were killed by asphyxiation using $CO_2$ gas. An incision in the lower abdomen exposed the male urogenital tract. The left and right prostate lobes were carefully dissected out for use in isolated organ bath studies. Excess fat and connective tissue were removed from tissues. Tissues were then placed in Krebs-Henseleit solution (NaCl 118.1 mM, KCl 4.7 mM, $MgSO_4.7H_2O$ 1.1 mM, $KH_2PO_4$ 1.2 mM, $NaHCO_3$ 25.0 mM, glucose 11.7 mM, $CaCl_2$ 2.5 mM).

## Isolated organ bath studies

Following dissection, rat prostate lobes were mounted in 10 ml water-jacketed glass organ baths containing Krebs-Henseleit solution. One end of the tissue was attached to a perspex tissue holder incorporating two vertical parallel platinum electrodes connected to a Grass S88 stimulator. The other end of the tissue was attached to an isometric Grass FT03 force displacement-transducer for the recording of smooth muscle contractions. Tissues were maintained at 37 ˚C and bubbled with 5% $CO_2$ in $O_2$ throughout the experiment. Developed force was recorded via a PowerLab data acquisition system (Chart 5.1) run on a personal computer. Prior to experimentation the tissues were equilibrated for 60 min under a resting force of 0.7–1.0 g. To ensure tissue viability, nerve terminals within the tissue were field stimulated during the equilibration period using electrical pulses of 0.5 ms duration and 60 V at 0.01 Hz. The organ bath medium was periodically replaced when secretions caused frothing to occur. When necessary the test extract or vehicle was replaced after bath washes. In all organ bath studies, isolated tissues treated with test extracts were compared to paired within animal control tissues treated with vehicle and run in parallel.

## Frequency-response curves

Following equilibration, frequency-response curves to electrical field stimulation (0.5 ms pulse duration, 60 V, 0.1–20 Hz) were constructed using a frequency progression ratio of approximately one third of a log unit. Trains of pulses were delivered at 10 min intervals. Each train consisted of 10 pulses for frequencies up to 1 Hz or trains of 10 s duration for frequencies greater than or equal to 1 Hz. At the completion of the frequency-response curve, the tissues were washed and allowed to rest for 30 min. A second frequency-response curve was performed after the tissue was exposed to the test sample or vehicle for a further period of 30 min.

## Exogenously administered agonists

In a separate set of experiments, following the equilibration period, noradrenaline (1 nM– 0.1 mM), acetylcholine (1 nM– 0.1 mM), adenosine 5'-triphosphate (ATP) (10 nM– 1 mM), or αβ-methylene ATP (3 nM– 10 μM) was used to construct discrete concentration-response curves on unstimulated tissues. Only one agonist was added to each tissue. A concentration progression ratio of half a log unit was employed. Once the contractile plateau had been reached, or if no response was observed after 20 seconds, the tissue was washed and allowed a 10 minute recovery period before the next concentration was applied. After a 30 min rest period, a second concentration-response curve was performed after the tissue was exposed to the test sample or vehicle for a further period of 30 min. This before and after treatment protocol alongside a parallel within animal time and vehicle control was used to monitor whether changes in tissue sensitivity occurred over the time course of the experimental design.

## Mouse mating studies

**Blood pressure and heart rate analyses.**   General cardiovascular health of male mice during mating studies was assessed by daily non-invasive measurement of cardiovascular parameters. Prior to measuring cardiovascular parameters, 6 eight-week-old male P2X1-purinoceptor knockout and 12 male wild-type mice were housed in a reverse light-cycle facility (12 hours light/dark; 7:00 am off, 7:00 pm on) for three days to acclimatise. Resting blood pressure (mmHg) and heart rate measurements (beats per minute (bpm)) were obtained using the tail cuff method with a non-invasive blood pressure analysis system (SC1000, Hatteras Instruments, Cary, NC, U.S.A.) connected to a personal computer, as previously described [27]. Over a period of five consecutive days, analysis comprised of 20 repeat blood pressure and heart rate measurements per day. After two days of rest, six wild type male mice were randomly allocated to the control group and orally dosed daily with vehicle (100 μl of 25% ethanol) and six wild type mice were randomly allocated to the treatment group and orally dosed with stinging nettle leaf extract (100 μl of 500 mg/ml) until the end of the mating study. The dose of extract was based on the manufacturer's maximum recommended daily dose for men (40 ml of 500 mg/ml) translated to an approximately equivalent dose for adult mice on a per kg basis with an allowance for the greater metabolism of mice. Daily measurement means were pooled to calculate mean and standard deviation for each mouse with final mean systolic blood pressure and heart rate values determined for each treatment group. P2X1-purinoceptor knockout male mice were not treated but cardiovascular parameters were measured over the same time course. The order of treatments and measurements of the different treatment groups was alternated daily to avoid confounding factors.

**Breeding observations.**   Subsequent to analysis of cardiovascular parameters, P2X1-purinoceptor knockout and treated wild-type male mice were mated with seven to eight-week-old female wild type mice to test their fertility. Wild-type female mice were housed in a reverse light-cycle facility (12 hours light/dark; 7:00 am off, 7:00 pm on) for three days to acclimatise. The female was then placed in the cage of the male in a dark behavioural room for two hours per day for up to nine days or until copulation to the point of ejaculation was confirmed. A B/W CCD camera (VIDO) and infrared lights were used to observe the mice, and video files were recorded using Windows movie maker (Microsoft) for record keeping and subsequent analysis. Once copulation had been confirmed, the female remained in her cage where she was allowed to gestate for 14 days before being sacrificed and examined to determine whether pregnancy had occurred, and the number of implanted foetuses were recorded. Each male was mated with two individual wild-type female mice. In total, 6 P2X1-purinoceptor knockout male mice, 12 wild type male mice and 36 wild type female mice were used in the breeding

study. Sample size was based on previous priori sample size calculations [27]. All animals used in the breeding study were included in the analysis.

**Mouse dissection.** Following breeding studies, male mice were weighed and then killed by asphyxiation using $CO_2$ gas. An incision was made along the midline of the abdomen. The prostate, vasa deferentia, bladder, testes, seminal vesicles, kidneys, liver, spleen, and heart were all carefully dissected out and weighed; the length of the vasa deferentia was measured. Female mice were killed using the above procedure. An incision was made along the midline of the abdomen to reveal the uterine horns. The presence or absence of foetuses were noted and foetal number counted and recorded.

## Statistical analysis

The number of experimental animals used is represented by *n*. In isolated organ bath experiments, the prostatic force (g) at the peak height of contraction, were measured in response to discrete frequencies of electrical field stimulation or concentrations of agonists, in the absence or presence of the test extract or vehicle. All data were analysed by two-way repeated measures analysis of variance (ANOVA) with Bonferroni post-test correction for multiple comparisons applied if required using GraphPad Prism version 5.0. The *p*-value for the interaction between treatment and frequency or treatment and agonist concentration was used to assess significance. This enabled a comparison of the differences between the control and treatment groups at all frequencies and concentrations on the frequency and concentration-response curve. Measurements from each tissue sample were pooled and expressed as the mean ± the standard error of the mean (SEM). *P*-values were used to represent the probability of the observed changes being due to chance. A value of $p < 0.05$ was considered statistically significant.

## Extracts and reagents

Stinging nettle root and leaf extracts (500 mg dried plant material/ml in 25% alcohol) were purchased commercially from MediHerb® Pty. Ltd. (Warwick, Queensland, Australia). Certified reference plant material was checked against the British Pharmacopoeia and authenticated by Mr Howard Hollow, Southern Cross University, Centre for Phytochemistry and Pharmacology (Lismore, NSW, Australia). The plant voucher specimen (reference material: MediHerb batch number T7C068; sample ID #CPA080318) is retained at Southern Cross University. These extracts are available in Australia for practitioner dispensing. Noradrenaline bitartrate salt (Arterenol®, Sigma) was dissolved and diluted to required concentrations using a catecholamine diluent (NaCl 154 mM, NaH2PO4 1.2 mM, ascorbic acid 0.2 mM in distilled water). Acetylcholine chloride (Sigma), ATP magnesium salt (Sigma), and αβ-methylene ATP lithium salt (Sigma) were all dissolved and diluted to required concentrations using distilled water. All drugs were made fresh on the morning of experimentation. Water used in all experiments was distilled using the MilliPore system (Burlington, MA, U.S.A.).

## Chemical separation by liquid-liquid partitioning

Initially undissolved solids in the sample were removed from stinging nettle leaf extract (100 ml) by vacuum filtration (Whatman filter paper 55) and ethanol vehicle evaporated off via rotavap. Aqueous residue was extracted with distilled ethyl acetate (30 ml) three times, and the resulting organic layers combined. Approximately 5 ml of saturated sodium chloride solution was added to the combined organic phase to remove leftover aqueous material. The resulting organic layer was then dried with magnesium sulfate, filtered and the solvent evaporated. The remaining aqueous extract was lyophilised to dryness.

## Results

### Effects of stinging nettle extracts on contractile responses to electrical field stimulation

Electrical field stimulation of the nerve terminals (60 V, 0.5 ms, 0.1–20 Hz for ten pulses (0.1–1 Hz) or 10 s (1–20 Hz) at 10 min intervals) elicited frequency-dependent contractions in isolated rat prostates in all experiments. Contractile responses of the isolated rat prostate gland to electrical field stimulation were consistent over the time course of the experiment ($p = 0.105$; n = 6) and were unaffected by vehicle ($p = 0.837$; n = 6). Incubation of isolated rat prostate glands in stinging nettle root extract (5–15 mg/ml) did not exert any effect on contractions in response to electrical field stimulation ($p \geq 0.135$; n = 6). On the other hand, stinging nettle leaf extract (5–15 mg/ml) produced a concentration related attenuation of electrical field stimulation-induced contraction ($p = 0.023$; Fig 1A & 1B).

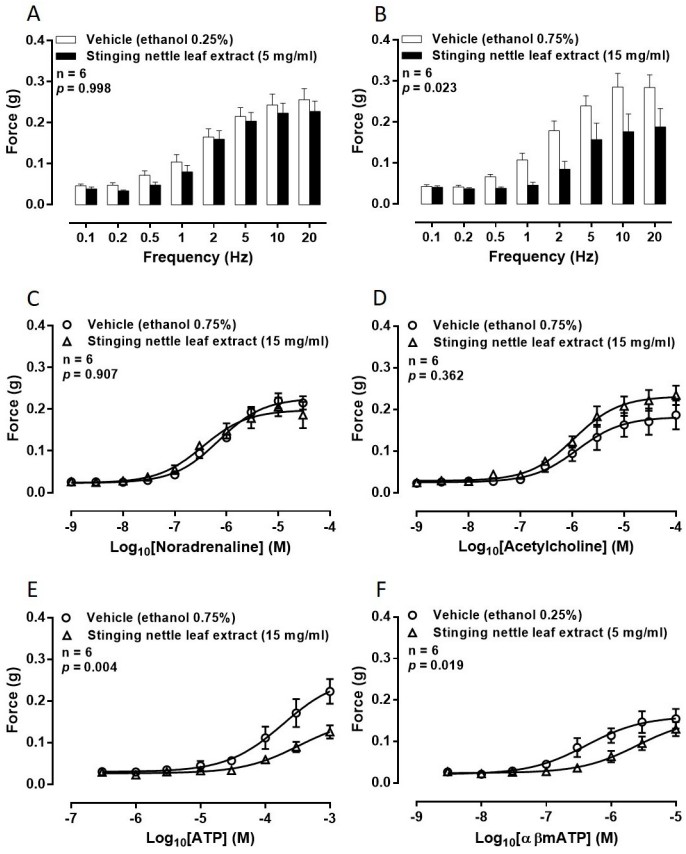

**Fig 1. Effect of stinging nettle leaf extract on prostate smooth muscle contractility.** Mean contractile responses to electrical field stimulation (0.5 ms, 60 V, 0.1 Hz to 20 Hz for 10 pulses or 10 seconds) of isolated rat prostate gland preparations in the presence of (A) vehicle (ethanol 0.25% v/v) and stinging nettle leaf extract (5 mg/ml) or (B) vehicle (ethanol 0.75% v/v) and stinging nettle leaf extract (15 mg/ml). Mean contractile responses to exogenously administered (C) noradrenaline (1 nM– 30 μM), (D) acetylcholine (1 nM– 100 μM), (E) adenosine 5'-triphosphate (ATP: 300 nM– 1 mM) and (F) α,β-methylene ATP (3 nM– 10 μM) of isolated rat prostate gland preparations in the presence of [○] vehicle (ethanol 0.25% or 0.75% v/v) and [□] stinging nettle leaf extract (5 or 15 mg/ml). In all graphs, each column or point represents the mean force generated by prostates taken from six rats. Error bars represent SEM. ANOVA *p*-values were determined by two-way repeated-measures ANOVA and represent the probability that the observed differences were due to chance.

## Effects of stinging nettle leaf extract on contractile responses to exogenously administered agonists

Concentration-dependent contractions of rat prostatic tissue were observed to each of the exogenously administered agonists. In addition, contractile responses of the isolated rat prostate gland to all of the exogenously administered agonists were consistent over the time course of the experiment and were unaffected by vehicle ($p \geq 0.744$; n = 6 for each agonist).

Noradrenaline-induced contractions were not attenuated in the presence of stinging nettle leaf extract at concentrations up to 15 mg/ml ($p = 0.907$; Fig 1C) when compared to paired vehicle controls. Similarly, maximum responses of the tissue were not different in the presence of stinging nettle leaf extract when compared to vehicle controls.

Stinging nettle leaf extract, at concentrations up to 15 mg/ml, did not attenuate acetylcholine-induced contractions ($p = 0.362$; Fig 1D) when compared to paired vehicle controls. Maximum responses of the tissue to acetylcholine were also not different in the presence of stinging nettle leaf extract when compared to vehicle controls. Similarly, there was no observable shift in the concentration-response curve over the time course in a vehicle control experiment ($p = 0.840$; n = 6) when acetylcholine was used to contract isolated rat prostate glands.

In contrast, as observed with electrical field stimulation-induced contraction, stinging nettle leaf extract (15 mg/ml) attenuated ATP-induced contraction in the isolated rat prostate gland ($p = 0.004$; Fig 1E) when compared to paired vehicle controls. To confirm whether the attenuation of ATP-induced contraction was due to the extract and not desensitisation of the P2X1-purinoceptors, these experiments were repeated using the more stable analogue αβ-methylene ATP. Attenuation of αβ-methylene ATP-induced contractions was observed in the presence of 5 mg/ml of stinging nettle leaf extract ($p = 0.019$; Fig 1F) when compared to paired vehicle controls.

## Liquid-liquid partitioning

Initially 50 ml of stinging nettle leaf extract (500 mg/ml) was filtered and separated into its organic and aqueous components yielding 59.02 mg and 3.49 g respectively. The organic fraction did not exhibit inhibitory activity ($p = 0.790$; Fig 2A) whereas the aqueous component was found to attenuate electrical field stimulation-induced contractions of isolated rat prostate gland ($p = 0.028$; Fig 2B). In both experiments there was no difference in the contractile responses to electrical field stimulation of the tissues prior to the administration of either vehicle or test compound ($p \geq 0.996$, n = 6).

To ascertain whether the bioactive or bioactives retained activity at the purinoceptors after partitioning, the aqueous fraction was tested on αβ-methylene ATP-induced contractions of the isolated rat prostate gland. A parallel shift to the right of the concentration-response curve to αβ-methylene ATP was apparent when the aqueous fraction was tested against αβ-methylene ATP ($p < 0.0001$; Fig 2C).

## Mouse mating fertility studies

Since the prostate and vas deferens of rodents are known to share the same P2X1-purinoceptor contractile mechanism, an *in vivo* fertility study was carried out to see whether *ex vivo* observations translated to subfertility in males.

## Behaviour

Normal libido was maintained with breeding observations showing normal sniffing, chasing, mounting and mating in all mice from all treatment groups with copulation occurring to the

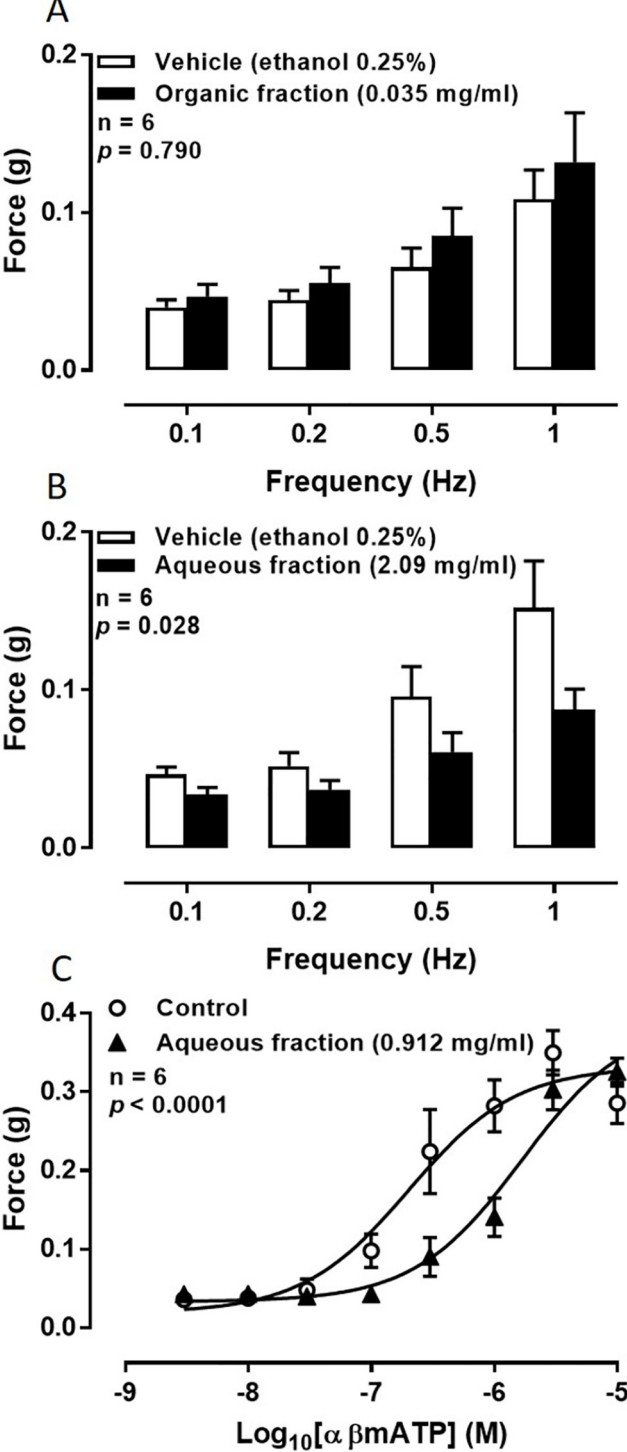

**Fig 2. Effect of aqueous and organic fractions of stinging nettle leaf extract on smooth muscle contractions.** Mean contractile responses to EFS (0.5 ms, 60 V, 0.1 Hz to 1 Hz for 10 pulses) of isolated rat prostate gland preparations in the presence of (A) vehicle (ethanol 0.25% v/v) and the organic fraction of stinging nettle leaf extract (0.035 mg/ml) or (B) vehicle (ethanol 0.25% v/v) and the aqueous fraction of stinging nettle leaf extract (2.09 mg/ml). Columns represent the mean force generated by prostates taken from six rats. Error bars represent SEM; and (C) mean contractile responses to exogenously administered α,β-methylene ATP (3 nM– 10 μM) of isolated rat prostate gland preparations in the presence of [○] vehicle (ethanol 0.25%) and [▲] the aqueous fraction of stinging nettle leaf extract (2.09 mg/ml, lower panel). Each point represents the mean force generated by prostates taken from six rats. Error bars represent

SEM. In all graphs *p*-values were determined by two-way repeated-measures ANOVA and represent the probability that the observed differences were due to chance.

point of ejaculation as observed by typical post-copulation behaviour of male mice (n = 6 male mice, each mated with two females).

## Effects of P2X1-purinoceptor receptor deletion and stinging nettle leaf extract on the cardiovascular system

Resting systolic blood pressure did not differ between P2X1-purinoceptor knockout and wild-type male mice at the beginning of the study (Fig 3A). Similarly, treatment with stinging nettle leaf extract or vehicle did not affect the systolic blood pressure of wild-type male mice over the five-day dosing period (*p* = 0.072; Fig 3A). In addition, the pulse rate did not differ between the P2X1-purinoceptor knockout and wild-type male mice, while the administration of stinging nettle leaf extract or vehicle, also did not alter pulse rate in wild-type mice over the five-day dosing period (*p* = 0.057; Fig 3B).

## Effect of P2X1-purinoceptor receptor deletion and stinging nettle leaf extract on body and organ weights

No differences were noted in the body weights among the P2X1-purinoceptor knockout and differently treated wild-type mice nor in the weights of their spleen, kidney, liver, or heart (Table 1). Within the lower urogenital tract, there were changes in the vas deferens, prostate, bladder and testes, but not in the seminal vesicles (Table 1). As previously observed, the length of the vasa deferentia was greater in P2X1-purinoceptor knockout mice compared to wild type mice but not different between wild type mice treated with stinging nettle leaf extract or vehicle (Table 1). Accordingly, there was also an increase in weight of the vas deferens taken from P2X1-purinoceptor knockout mice when compared to vehicle-treated wild-type mice (Table 1), but no difference in vas deferens weight between the vehicle-treated and stinging nettle leaf extract-treated wild-type mice (Table 1). Prostate glands were observed to be heavier in the P2X1-purinoceptor knockout mice when compared to both the vehicle-treated and the stinging nettle leaf extract-treated wild-type mice (Table 1). Differences were observed in the weights of both the testes and the bladder between treatment groups, with stinging nettle leaf extract-treated mice presenting with slightly lower testis and bladder weights than vehicle treated mice and P2X1-purinoceptor knockout mice having slightly heavier testis and bladder weights (Table 1).

## Effect of P2X1-purinoceptor receptor deletion and stinging nettle leaf extract on fertility of male mice

Following gestation, the positive pregnancy rate and foetal number was determined for each wild-type female mate to ascertain the effect of either genotype, or treatment with either stinging nettle leaf extract or vehicle on fertility in male mice. Both the pregnancy rate and the mean foetal number in females mated with males treated with stinging nettle leaf extract or P2X1-KO male mice, were reduced. Of the twelve wild-type females mated with P2X1-purinoceptor KO male mice only one pregnancy occurred (8.3%; mean foetal number = 0.25 ± 0.35 per mating). For stinging nettle leaf extract treated wild-type males, two males failed to produce pregnancies with either female mate, while two males impregnated one of their female mates and the two other males impregnated both female mates. Overall, pregnancy occurred

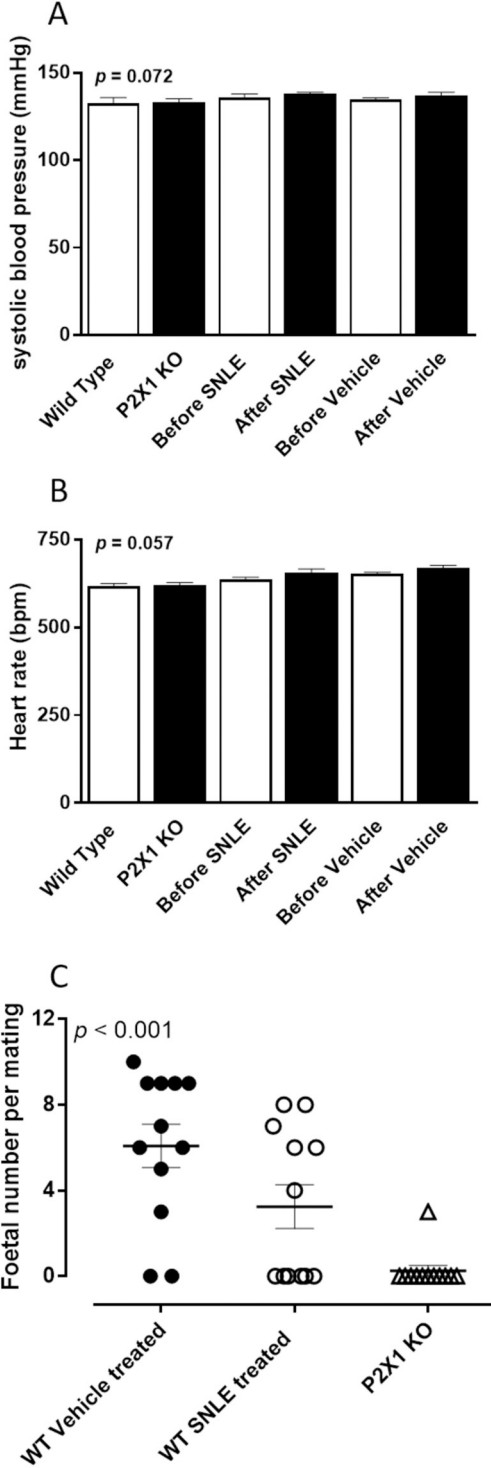

**Fig 3. Effect of stinging nettle leaf extract or P2X1-purinoceptor deletion on resting cardiovascular parameters and male fertility.** Mean resting (A) systolic blood pressure (mmHg) and (B) heart rate (bpm) measured by the tail cuff method in untreated age matched wild-type (n = 6) and P2X1-purinoceptor knockout (KO) (n = 6), and wild-type male mice, before and after daily oral dosing with stinging nettle leaf extract (SNLE) (100 μl of 500 mg/ml) or vehicle (100 μl of 20% ethanol), where n = 6 for each treatment variable. Columns represent the mean systolic blood pressure (A) or heart rate (B) over 5 consecutive days prior to the commencement of treatment or 5 consecutive days after the commencement of treatment. Error bars represent SEM. ANOVA *p*-values were determined by two-way repeated-measures ANOVA and represent the probability of genotype or treatment causing a significant change in systolic

blood pressure or heart rate. (C) Foetal number resulting from wild-type female mice matings with wild-type male mice treated with vehicle (100 μl of 25% ethanol) (●, an oral daily dosing of stinging nettle leaf extract (SNLE) (100 μl of 500 mg/ml) (○) or age matched P2X1-purinoceptor knockout male mice (Δ). Six male mice were each mated with two female wild type mice for each treatment variable. The centre line represents the mean foetal number. Error bars represent SEM. ANOVA *p*-values were determined by one-way ANOVA and represent the probability of genotype or treatment affecting male fertility represented by the resultant foetal number.

in six of the twelve mated wild-type females (50%; mean foetal number = 3.25 ± 1.43 per mating), whereas 10 out of the 12 mated wild-type females were found to be pregnant (83%; mean foetal number 6.09 ± 1.45 per mating) after mating with vehicle treated wild-type male mice (Fig 3C). Two of the vehicle treated males only produced pregnancies with one of their female mates.

## Discussion

Stinging nettle leaf extract demonstrated an ability to reduce the contractility of both electrical field stimulation induced contraction as well as ATP and αβmethylene-ATP mediated contraction. These results suggest that the extract works postjunctionally, most likely as a P2X1-purinoceptor antagonist. Although the selectivity of stinging nettle leaf extract for P2X1 over other subtypes of P2-purinoceptor was not investigated in this study, contractions of the prostate and vas deferens elicited by ATP are known to be mediated by the P2X1 subtype in mice, rats and guinea-pigs [27, 29–31]. Therefore, we propose or have assumed that P2X1 is the purinoceptor subtype being affected in this case. Nevertheless, various isoforms of P2X-purinoceptors exist in the testis and other systems, so it is possible that stinging nettle leaf extract may have had some non-specific effects at these sites to produce its subfertility effects.

Interestingly, oral treatment with stinging nettle leaf extract resulted in reduced weight of the testis and bladder in treated mice, while P2X1 knockout mice had increased weights of these tissues. This discrepancy was unexpected but several factors may contribute to this confounding observation. Firstly, plant extracts contain multiple biochemical components and any number of these may activate biological mechanisms distinct from purinoceptors that may

**Table 1. Mean body and tissue weights or lengths in age-matched vehicle and stinging nettle leaf extract (SNLE) treated wild-type and P2X1-purinoceptor knockout (KO) mice.**

| | Wild-type (Mean ± SEM) | | P2X1-KO (Mean ± SEM) |
|---|---|---|---|
| | **Vehicle treated** | **SNLE treated** | |
| **Body weight (g)** | 26.3 ± 0.5 | 25.7 ± 0.6 | 27.2 ± 0.9 |
| **Vas deferens length (mm)** | 24.4 ± 0.2 | 24.3 ± 0.2 | 28.6 ± 0.1* |
| **Vas deferens weight (mg)** | 11.5 ± 1.0 | 12.5 ± 1.1 | 16.9 ± 1.2* |
| **Prostate weight (mg)** | 16.7 ± 2.2 | 17.5 ± 2.4 | 29.3 ± 1.9** |
| **Seminal vesicle weight (mg)** | 50.6 ± 3.6 | 54.5 ± 6.3 | 50.4 ± 2.2 |
| **Testis weight (mg)** | 97.5 ± 2.3 | 89.3 ± 2.3* | 102.3 ± 1.7* |
| **Bladder weight (mg)** | 31.8 ± 2.0 | 26.8 ± 1.4* | 43.3 ± 3.4* |
| **Kidney weight (mg)** | 193.0 ± 6.2 | 197.7 ± 9.2 | 201.0 ± 9.6 |
| **Spleen weight (mg)** | 66.7 ± 5.9 | 71.8 ± 7.1 | 87.3 ± 7.3 |
| **Liver weight (g)** | 1.25 ± 0.06 | 1.22 ± 0.09 | 1.24 ± 0.08 |
| **Heart weight (mg)** | 131.4 ± 6.3 | 132.0 ± 6.6 | 145.9 ± 11.2 |

* $p < 0.05$ and ** $p < 0.001$ represent the probability of a significant difference in the length or weight, from vehicle-treated wild-type mice. All *p* values were calculated by a one-way ANOVA test with a Bonferroni correction for multiple comparisons (n = 12 for each measurement).

have contributed to this discrepancy. Secondly, the genetically modified mice would lack P2X1-purinoceptors from conception, meaning that they lacked P2X1-purinoceptors throughout their entire development and not just for the timeframe of our study as in the case of the stinging nettle leaf extract treated mice. Not only could this lifelong change affect tissue development directly but it would also allow for compensatory biological mechanisms to develop that may affect the weights of tissues, as observed.

Although the root extract of stinging nettle is widely used for urinary complications in traditional medicine, it did not exert an effect on the contractility of the prostate. This does not necessarily mean that the root extract is not beneficial in the treatment of BPH as it may still be slowing growth of the prostate to provide symptomatic benefits rather than affecting the contractility of the prostate gland.

Conventional medical treatments of LUTS associated with BPH act by either inhibiting $5\alpha$-reductase enzymes decreasing hyperplasia or using $\alpha_{1A}$-adrenoceptor antagonists to reduce the contractile tone of smooth muscle within the gland. Previous studies have demonstrated a residual contraction to nerve stimulation in both the guinea pig and rat prostate after $\alpha_1$-adrenoceptor antagonists have been administered [29, 30]. This residual contraction in both species was attributed to a purinergic mechanism via neuronally released ATP activating P2X1-purinoceptors [30, 31]. Targeting both $\alpha_{1A}$-adrenoceptors as well as P2X1-purinoceptors may potentially reduce the contractility of the prostate gland to a greater extent than $\alpha_{1A}$-adrenoceptor antagonism alone and may therefore result in greater symptom control when treating BPH. There may also be the possibility of reducing the doses by using a combination treatment, which may allow a reduction in side effects, however unexpected adverse drug interactions may arise due to the use of a combination treatment.

It is possible that stinging nettle leaf extract may potentially be acting to inhibit a downstream process to P2X activation instead of as a receptor antagonist. However, this is unlikely because P2X1-purinoceptors are ATP ligand gated ion channels that when activated by ATP or its analogues directly open an ionic pore that allows extracellular $Ca^{2+}$ ions to flow into the smooth muscle cell cytoplasm. This influx of $Ca^{2+}$ ions interacts with the smooth muscle machinery directly to produce contraction. An increase in cytoplasmic $Ca^{2+}$ ions is necessary to produce muscle contraction, therefore any downstream inhibitory effects would also inhibit contractile responses to stimulation of other mechanisms such as noradrenergic or cholinergic receptors. This was not observed, implying that inhibition of P2X-purinoceptors is the most likely mechanism of action. Furthermore, it has been established and accepted through observations using immunohistochemical and pharmacological techniques that the only purinoceptor that mediates contractions of the rat prostate smooth muscle is of the P2X1-subtype [30]. Currently, known P2X1-purinoceptor antagonists are not selective. Furthermore, antagonists such as suramin and its derivatives are large polysulfonated molecules that have physical and chemical properties that are not suitable for progression as drug candidates [32]. Consequently, derivatives of suramin and most other P2X1-purinoceptor antagonists exist only as pharmacological tools. The discovery of an orally bioavailable P2X1-purinoceptor selective compound would be beneficial in future, more effective treatments for BPH. The purinoceptor subtype selectivity of stinging nettle leaf extract was not investigated during this study, and considering that botanical extracts contain a multitude of compounds, stinging nettle leaf extract itself may be promiscuous across various purinergic subtypes. However, the isolation of a single active component may prove to be a good lead candidate for development of P2X1-purinoceptor antagonists.

The mating study described here, demonstrated that pharmacological blockade of the P2X1-purinoceptor, by stinging nettle leaf extract, reduced male fertility, when compared to untreated wild-type male mice, using a daily oral administration regime. Although the degree

of infertility was less than observed in P2X1-purinoceptor knockout mice, a higher dose of stinging nettle leaf extract may have led to a more similar degree of reduction in male fertility. The identification, extraction, and purification of the active compound may further improve efficacy and selectivity. Nevertheless, this study demonstrates that oral administration of a P2X1-purinoceptor antagonist is able to reduce male fertility in mice and therefore further validates the proposal that pharmacological blockade of $\alpha_{1A}$-adrenoceptors and P2X1-purinoceptors is a viable therapeutic strategy for a nonhormonal oral male contraceptive [27].

P2X1-purinoceptors, alongside $\alpha_{1A}$-adrenoceptors, are known to be responsible for the contractility of the vas deferens and subsequently the transport of sperm from the cauda epididymis to the base of the urethra during the ejaculation process. Previously shown in two separate studies, the knockout of P2X1 purinoceptors [33] as well as dual $\alpha_{1A}$-adrenoceptor / P2X1-purinoceptor knockout in male mice [27], results in a significant reduction in the fertility of male mice, without detrimental effects on male characteristics or sperm viability. The reduction in the contractility of the vas deferens impedes the transport of sperm during the ejaculation process, resulting in a reduced rate of pregnancy. As the mice in both of these studies were seen to be phenotypically normal, it has been proposed that the use of selective antagonists at these receptors to induce male contraception would be well tolerated.

Safe and effective pharmacological antagonists of $\alpha_1$-adrenoceptors are currently available, and used clinically for the treatment of BPH and hypertension. However the $\alpha$-adrenoceptor antagonist used and the route of administration affects the presence of spermatozoa in the ejaculate as well as ejaculation function. Prazosin delivered locally to the vas deferens or epididymis of healthy adult male rats, by way of silastic formulations in the form of rods or collars, reduced fertility with no effect on mating or courting behaviour, or on the motility of spermatozoa [34]. Conversely, oral prazosin was found to have no effect on healthy adult human male fertility [35]. Selective therapeutic blockade of $\alpha_{1A}$-adrenoceptors with tamsulosin has also been shown to inhibit the fertility of male rats [36]. At normal doses used in the symptomatic treatment of BPH, tamsulosin significantly reduced total functional sperm count in semen in healthy men with an associated marked decrease in ejaculate volume and occasional anejaculation [37–39]. However, at therapeutic doses, another selective $\alpha_{1A}$-adrenoceptor antagonist, alfuzosin, did not decrease total sperm count or cause ejaculation dysfunction [38, 40]. Differences in the response of the epididymal and prostatic portions of the human vas deferens to exogenous noradrenaline, as well as differences in the activity and selectivity of various $\alpha_1$-adrenoceptor antagonists on the two distinct segments, has been suggested as a possible reason to explain these differences [41]. The introduction of a P2X1-purinoceptor antagonist along with an effective $\alpha_1$-adrenoceptor antagonist may augment the efficacy and reduce side effects associated with doses required to achieve male contraception. This study therefore identifies that aqueous stinging nettle leaf extract contains a component that is an orally viable P2X1-purinoceptor antagonist. Once isolated and identified, this may have potential as a drug-like lead compound that enables the synthesis of an orally active, potent small molecule compound for use in combination with tamsulosin for the development of a male contraceptive.

## Author Contributions

**Conceptualization:** Jamie S. Simpson, Sabatino Ventura.

**Data curation:** Nicole T. Eise.

**Formal analysis:** Nicole T. Eise, Jamie S. Simpson, Philip E. Thompson, Sabatino Ventura.

**Investigation:** Nicole T. Eise, Philip E. Thompson.

**Methodology:** Nicole T. Eise, Jamie S. Simpson, Philip E. Thompson, Sabatino Ventura.

**Project administration:** Jamie S. Simpson, Philip E. Thompson, Sabatino Ventura.

**Resources:** Jamie S. Simpson, Philip E. Thompson, Sabatino Ventura.

**Supervision:** Jamie S. Simpson, Philip E. Thompson, Sabatino Ventura.

**Validation:** Sabatino Ventura.

**Writing – original draft:** Nicole T. Eise.

**Writing – review & editing:** Jamie S. Simpson, Philip E. Thompson, Sabatino Ventura.

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
