## [Decision Letter · Decision Letter 0]

23 Jun 2022

PONE-D-22-15650Aqueous extracts of Urtica dioica (stinging nettle) leaf contain a P2X1-purinoceptor antagonist – implications for male fertilityPLOS ONE

Dear Dr. Ventura,

Thank you for submitting your manuscript to PLOS ONE. After careful consideration, we feel that it has merit but does not fully meet PLOS ONE’s publication criteria as it currently stands. Therefore, we invite you to submit a revised version of the manuscript that addresses the points raised during the review process. Specifically, a longer interval between stimuli and some direct finding that indicates the presence of a P2X1 pharmacological antagonist are important issues that should be considered in the revised version. 

We look forward to receiving your revised manuscript.

Kind regards,

Luis Eduardo M Quintas, Ph.D.

Academic Editor

PLOS ONE

Journal Requirements:

2. Please amend the manuscript submission data (via Edit Submission) to include author  Philip E. Thompson.

Reviewers' comments:

Reviewer's Responses to Questions

**Comments to the Author**

1. Is the manuscript technically sound, and do the data support the conclusions?

Reviewer #1: Yes

Reviewer #2: Partly

2. Has the statistical analysis been performed appropriately and rigorously? 

Reviewer #1: Yes

Reviewer #2: Yes

3. Have the authors made all data underlying the findings in their manuscript fully available?

Reviewer #1: Yes

Reviewer #2: Yes

4. Is the manuscript presented in an intelligible fashion and written in standard English?

Reviewer #1: Yes

Reviewer #2: Yes

5. Review Comments to the Author

Reviewer #1: Title: Aqueous extracts of Urtica dioica (stinging nettle) leaf contain a P2X1-purinoceptor

antagonist – implications for male fertility

Overview: This manuscript reports that stinging nettle leaf extract (SNLE) attenuates electrical field stimulation induced rat prostate smooth muscle contractions and reduces fertility in male mice. The authors report that SNLE attenuates ATP- and αβmATP-induced rat prostate contractions, but not acetylcholine- or noradrenaline-induced contractions. As a result, they propose that SNLE is functioning as a P2X1 purinergic receptor antagonist, based on previous evidence that P2X1 receptors are involved in residual contraction of prostate. This manuscript is novel in its presentation of SNLE as a mediator of male infertility and corroborates previous studies identifying P2X1 as a potential target for male contraception. The data presented will be of interest to the P2 purinergic receptor and the male fertility readership. With minor revisions this manuscript is a welcomed addition to the literature.

Comments:

1. The abstract states that SN extracts were tested for their effect on prostatic and vas deferens smooth muscle contractility. However, neither the figures or text present data on vas deferens contractility, only weight.

2. The authors acknowledge in the discussion that the selectivity of SNLE was not investigated and therefore any other P2 purinergic receptor may be involved. This is appreciated. However, the language throughout, including the title, is quite definitive about P2X1. Perhaps in the first paragraph of the discussion, that P2X1 is a ‘proposed’ or ‘assumed’ receptor should be more explicit. Similarly, reconsider the definitive stating of P2X1 in the title, perhaps using a qualifier.

3. In Table 1, treatment with SNLE resulted in reduced weight of the testis and bladder, however P2X1 KO mice had increased weights of these tissues. This discrepancy is not discussed. Perhaps the authors may include a section in the discussion that addresses this.

4. Fig. 3A,B: The current labelling is not immediately clear. For the KO what is the before and after? Perhaps label the remaining four as WT? Consider the labelling below.

'P2X4 KO' 'WT'

Veh. - - - - - +

SNLE - + - + - -

Reviewer #2: Major revisions:

- the authors present vehicle vs. leaf extract data. In the methods section it is mentioned that they acquired contraction data from nonstimulated prostate preparations before the stimulation with either vehicle or leaf extract. I suggest including the nonstimulated control data, enabling the reader to assess the presence or absence of a vehicle artifact.

-P2X1 is a fast-desensitizing, highly ATP-sensitive ion channel. Given the short inter-stimulus-interval (ISI) of only 10 min (Fig. 1 E,F & Fig. 2 C) I'm concerned that the channel will still be somewhat desensitized (see: Rettinger & Schmalzing (2003)). This might explain why such high ATP concentrations are required to elicit measurable contractions when the EC50 of P2X1 is just 700 nM (Rettinger & Schmalzing (2003)). It is also not clear to me how the use of alpha-beta-methylene ATP should prevent the channel from desensitizing (lines 312-314). The authors should show that a longer ISI of >30 min results in comparable ATP-induced contractions under control conditions.

- the authors claim that the leaf extract contains a P2X1 antagonist. At this point their data does not support this claim. Direct physiologcal evidence (i.e. calcium imaging, electrophysiology of (recombinant) P2X1) is required to support this claim. The leaf extract could potentially inhibit downstream processes of P2X activation instead of acting as a bona fide P2X1 antagonist. The authors should either support their claim with the missing physiological data or change the narrative of their manuscript.

-Line 26: "Stinging nettle root and leaf extracts were tested for their effect on prostatic and vas deferens

smooth muscle contractility". I can only find data obtained from prostate preparations. To link their findings with the observed subfertility effect, the authors should investigate whether leaf extract can also inhibit ATP-induced vas deferens contractions.

-Figure 3C: please address how many males contributed to the resulting pregnancies of the treated and nontreated groups

Minor revisions:

-Since various P2X isoforms are expressed in the testis and other male reproductive subsystems, it would be interesting to see testis histology and/or sperm counts of leaf extract treated male mice. It's unclear how P2X1-specific the putative leaf extract antagonist is and what side effects it might elicit by acting on other P2X isoforms. This would also be relevant for the application as a male contraception.

-Line 340: Heading seems to be misplaced. Should be moved to line 347

- discrepancy in author list: Philip Thompson does not appear in authors listed by PLOS ONE

6. PLOS authors have the option to publish the peer review history of their article (what does this mean?). If published, this will include your full peer review and any attached files.

Reviewer #1: **Yes: **Janielle P. Maynard

Reviewer #2: **Yes: **Nadine Mundt

---

## [Author Response · Author response to Decision Letter 0]

28 Jun 2022

We have responded to the comments of the Reviewers as follows:

Reviewer #1:

Reviewer 1 believed the manuscript to be novel as well as corroborating previous studies identifying P2X1 as a potential target for male contraception. They believed that the data presented would be of interest to readers in the field of P2 purinergic receptors and male fertility and a welcome addition to the literature. We have responded to their comments as follows:

1. Reviewer 1 points out that although stated in the abstract that SN extracts were tested for their effect on prostatic and vas deferens smooth muscle contractility, no data was presented on vas deferens contractility. We have now removed “and vas deferens” from this sentence to avoid this ambiguity (page 2).

2. The Reviewer indicates that we have not definitively shown that the receptor being modulated is of the P2X1 subtype and may be any other P2 purinergic receptor. We agree with the Reviewer but seeing as it is well known in the literature that contractions of the prostate and vas deferens elicited by ATP are mediated by the P2X1 subtype, we have assumed P2X1 to be the receptor being affected in this case. Nevertheless, in accordance with the Reviewer’s suggestion, we have modified the manuscript as follows:

• The title has been changed to qualify the nature of antagonism from “P2X1” to just “P2” (page 1).

• Two sentences explaining this concept have been incorporated into the opening paragraph of the Discussion (page 18).

3. As the Reviewer points out, SNLE treatment resulted in reduced testis and bladder weights, while P2X1 KO mice had increased weights of these tissues. This anomaly is difficult to explain but we have now addressed this discrepancy in our discussion (page 19).

4. The labelling of figure 3 and the corresponding caption have been amended in accordance with Reviewer 1’s comments.

Reviewer #2:

Reviewer 2 believed the manuscript required some major and minor revisions. We have responded to their comments as follows:

Major revisions:

• Reviewer 2 wishes us to include control data prior to the administration of vehicle or leaf extract to allow the reader to assess the presence or absence of a vehicle effect. Rather than show all of this data, we have stated for all ex vivo tissue experiments that contractions elicited by electrical field stimulation or exogenous administration of agonists “were unaffected by vehicle” implying that the magnitude of contractions was not different before or after incubation with vehicle. This implies that there was no vehicle artefact. A “p value” determined using repeated measures ANOVA is included with each statement so that the reader can assess the degree of statistical significance of any vehicle effect. A statement along these lines was already present in the Results section of the manuscript for the electrical field stimulation experiments (page 12) and we have now added another for the exogenously added agonist experiments (also page 12). We have not revised the manuscript further as we believe that the inclusion of these statements addresses this comment and the addition of five extra graphs would be required to show all of the data. We feel that inclusion of such a large amount of extra data may distract readers from the main message of our paper. However, we are happy to add these graphs as supplementary material if Reviewer 2 feels strongly about including them.

• Reviewer 2 expresses concern about the fast desensitizing nature of P2X1 receptors and suggests that a longer inter-stimulus interval should be tried. This is a valid concern, but we have vast experience with the use of ATP and its analogues to elicit contractile responses of isolated tissues. The choice of inter-stimulus interval chosen in these types of experiments are the result of a balance between the need to avoid desensitisation and the length of time that isolated tissues remain viable in an ex-vivo environment. Over a long period of time (>20 years), we have observed that isolated preparations of rat prostate yield reliably consistent concentration-related contractile responses for approximately 4-6 hours before their viability deteriorates. Consequently, due to the tissue equilibration and drug incubation times needed in our experimental design, we have settled on the 10 min inter-stimulus interval so that the experimental protocol can be completed safely within the tissue viability life span. We also perform laboratory protocols that avoid desensitisation such as washing tissues quickly after a peak contraction is reached and extra tissue washes. Furthermore, we have controlled for any change in tissue sensitivity due to receptor desensitisation by using a before and after treatment protocol alongside a parallel within animal tissue control. As stated in the Results section, parallel tissues produced contractions of similar magnitude and concentration-response curves within tissues did not change over the time course of the experiment following vehicle treatment of tissues. We believe that these observations confirm that receptor desensitisation does not diminish the sensitivity of tissues to ATP or its analogues over the time course of our experiments. A sentence to describe the rationale for these controls has been added to the Methods section (page 8). Using a longer inter-stimulus interval of > 30 min would not allow us to test sufficient concentrations of agonist in our experiment to cover an adequate concentration-response range for proper data analysis. Reviewer 2 also requests clarification on why α,β methylene-ATP should prevent desensitisation. α,β methylene-ATP is a more stable analogue of ATP which causes desensitisation of P2X1-purinoceptors more readily and our use of this agonist was to demonstrate that desensitisation did not occur using our experimental protocol as could be seen with vehicle treatment. No change was made to the manuscript.

• Reviewer 2 suggests that the stinging nettle leaf extract may potentially be acting to inhibit a downstream process to P2X activation instead of as a receptor antagonist. This is again a valid point. However, it is unlikely that the extract is inhibiting a downstream process because P2X1 receptors are ATP ligand gated ion channels that when activated by ATP or one of its analogues open an ionic pore that allows extracellular Ca2+ ions to flow into the smooth muscle cell cytoplasm. This influx of Ca2+ ions interacts with the smooth muscle machinery directly to produce contraction. An increase in cytoplasmic Ca2+ ions is necessary to produce muscle contraction, therefore any downstream inhibitory effects would also inhibit contractile responses to stimulation of other mechanisms such as noradrenergic or cholinergic receptors. This was not observed in our study implying that inhibition of P2X-purinoceptors was the mechanism of action. Furthermore, Reviewer 2 also suggests that direct physiological evidence such as Ca2+ imaging or electrophysiology is required to support the claim that the purinoceptor being inhibited is of the P2X1-subtype. However, we believe this not be necessary in this case as it has been largely accepted for many years through immunohistochemical and pharmacological techniques that the only purinoceptor that mediates contractions of the rat prostate smooth muscle is of the P2X1-subtype (Ventura et al, 2003; Br J Pharmacol: 138: 1277-1284). Several sentences have now been incorporated into the Discussion on these points (page 20).

• As Reviewer 1 also mentioned, Reviewer 2 points out that although stated in the abstract that SN extracts were tested for their effect on prostatic and vas deferens smooth muscle contractility, no data was presented on vas deferens contractility. We have now removed “and vas deferens” from this sentence to avoid this ambiguity. Nevertheless, from countless previously published isolated tissue studies, the prostate, vas deferens (and seminal vesicle) of rodents are known to share the same P2X1-purinoceptor contractile mechanism. In order to link our ex vivo findings to our fertility study, a sentence has been added to the Results section on page 15.

• Reviewer 2 asked how many males contributed to the resulting pregnancies in figure 3C. This data has now been incorporated into the text in the Results section (page 18).

Minor revisions:

• We agree with Reviewer 2 that various P2X isoforms are expressed in the testis and other male reproductive systems but histological testis examination and sperm counts following stinging nettle leaf extract is a large undertaking that is beyond the scope of this study. Nevertheless, we have added a comment in the Discussion section related to the possibility that leaf extract may be acting at these sites (page 19).

• The Reviewer is correct that this heading is misplaced and it has been moved to the correct position (page 14, line 352) as pointed out by the Reviewer. We thank Reviewer 2 for pointing out this error on our part.

• The discrepancy in the author list has been amended.

We thank the two Reviewers for their insightful and intuitive comments as we feel that the changes we have made in response to their comments have enhanced our manuscript.

---

## [Decision Letter · Decision Letter 1]

7 Jul 2022

Aqueous extracts of Urtica dioica (stinging nettle) leaf contain a P2X1-purinoceptor antagonist – implications for male fertility

PONE-D-22-15650R1

Dear Dr. Ventura,

We’re pleased to inform you that your manuscript has been judged scientifically suitable for publication and will be formally accepted for publication once it meets all outstanding technical requirements.

Kind regards,

Luis Eduardo M Quintas, Ph.D.

Academic Editor

PLOS ONE

Additional Editor Comments (optional):

Reviewers' comments:

Reviewer's Responses to Questions

**Comments to the Author**

1. If the authors have adequately addressed your comments raised in a previous round of review and you feel that this manuscript is now acceptable for publication, you may indicate that here to bypass the “Comments to the Author” section, enter your conflict of interest statement in the “Confidential to Editor” section, and submit your "Accept" recommendation.

Reviewer #1: All comments have been addressed

Reviewer #2: All comments have been addressed

2. Is the manuscript technically sound, and do the data support the conclusions?

Reviewer #1: Yes

Reviewer #2: Yes

3. Has the statistical analysis been performed appropriately and rigorously? 

Reviewer #1: Yes

Reviewer #2: Yes

4. Have the authors made all data underlying the findings in their manuscript fully available?

Reviewer #1: Yes

Reviewer #2: Yes

5. Is the manuscript presented in an intelligible fashion and written in standard English?

Reviewer #1: Yes

Reviewer #2: Yes

6. Review Comments to the Author

Reviewer #1: (No Response)

Reviewer #2: The authors have addressed my comments completely and in a reasonable manner. They have explained that the 10 minutes ISI is a compromise between desensitization minimization and longevity of ex vivo tissue preparations. In addition, concentration-response curves within tissues did not change over the time course of the experiment following vehicle treatment of tissues, which convinces me that desensitization is not an issue.

The authors also reason why the observed effect is most likely an antagonistic effect on P2X1 and the changes that the authors have made to the manuscript emphasize the possibility of non-P2X1 targets.

The absence of a vehicle effect is now mentioned in the manuscript and supported by the corresponding p-values.

My recommendation is, therefore, to accept the revised manuscript for publication.

7. PLOS authors have the option to publish the peer review history of their article (what does this mean?). If published, this will include your full peer review and any attached files.

Reviewer #1: **Yes: **Janielle Maynard

Reviewer #2: **Yes: **Nadine Mundt

---

## [Editor Report · Acceptance letter]

11 Jul 2022

PONE-D-22-15650R1 

Aqueous extracts of *Urtica dioica* (stinging nettle) leaf contain a P2-purinoceptor antagonist – implications for male fertility 

Dear Dr. Ventura:

I'm pleased to inform you that your manuscript has been deemed suitable for publication in PLOS ONE. Congratulations! Your manuscript is now with our production department. 

Kind regards, 

on behalf of

Dr. Luis Eduardo M Quintas 

Academic Editor

PLOS ONE